# Peer review of "Weak Ergodicity Breaking in Non-Hermitian Many-body Systems"

_SciPost Physics_

## Round 1 · Referee Report · Anonymous (Referee 1) · 2022-10-24

Strengths

1- thorough
2- easy to read

Weaknesses

1-not clear how to realize jump operators

Report

The authors study a non-Hermitian variant of the PXP model. They find that the weak ergodicity breaking (or more precisely long-lived oscillations) only exists in the region of the phase diagram that has real energies for the scarred states, which makes sense. The phase diagram is determined numerically in line with most studies on the PXP model.

The authors are further very thorough and study the spectral statistics, fidelity from special initial states, a few different prototypical observables (known from the Hermitian case to exist long-time oscillations), entanglement, etc.

In my opinion, importantly, the authors manage to connect their non-Hermitian Hamiltonian to a Lindblad master equation, which renders the setup more physical (though it remains unclear how to realize the required jump operators experimentally).

Why their model has real energies for the scars and why in that region remains unclear. I cannot find an intuitive picture, nor some sort of analytical argument.

Requested changes

1- Comment on possible experimental realizations of jump operators

2- The reason for the weak ergodicity breaking in region (I) could possible be understood through Supp Th 3 of Ref. [98] that connects scars in Lindblad master equations to purely imaginary eigenvalues of the Liouvillian and with pure state eigenmodes of the Liouvillian. I suggest that the authors maybe check this.

  • validity: high
  • significance: good
  • originality: good
  • clarity: top
  • formatting: excellent
  • grammar: excellent

Author:  Qianqian Chen  on 2023-02-28  [id 3412]

(in reply to Report 1 on 2022-10-24)

Response to Anonymous Report 1 on 2022-10-24 (Invited Report)

We would like to thank the referee for reviewing our manuscript. The constructive suggestions of the referee are of great help for us to revise our manuscript. We have accepted all of the “request changes” and revised the manuscript accordingly. We sincerely hope that these improvements and the appended point-to-point responses could resolve the referee’s concerns in a satisfactory manner.

Requested changes:

1- Comment on possible experimental realizations of jump operators

Our reply: The jump operators we constructed based on a perturbative analysis are complicated and it may be hard to realize such jump operators in experiments.

In our manuscript, what we identified is a condition for the jump operators that correspond to our non-Hermitian model, and there should be many other choices for such jump operators. Based on a perturbative analysis, we gave one of the examples of such jump operators so as to perform the numerical calculation. Future construction of experimentally realizable jump operators is fundamentally important to deepen our understanding of the QMBS in an open quantum system. We have highlighted these points in Section 3.5, the appendix, and the section “Summary and Discussion” of the revised manuscript.

2- The reason for the weak ergodicity breaking in region (I) could possibly be understood through Supp Th 3 of Ref. [98] that connects scars in Lindblad master equations to purely imaginary eigenvalues of the Liouvillian and with pure state eigenmode of the Liouvillian. I suggest that the authors maybe check this.

Our reply: We thank the referee for such constructive suggestion and for drawing our attention to Supp Th 3 of Ref. [98], which we believe is very significant for the mechanism of the weak ergodicity breaking (or more precisely, non-stationarity) in open quantum systems. However, for our model, we have found that the QMBS states are not the pure state eigenmodes of the Liouvillian we constructed. One may design other forms of Lindblad operators that satisfy the theorem, which is beyond the scope of the current work. Indeed, in both the Hermitian PXP model and the non-Hermitian PXP model we studied, the QMBS states are not exactly commensurately spaced in energy, which may lead to relaxation. In addition, the mechanisms causing scars in the PXP model are only approximately understood, usually with some deformations, while here we did not consider those deformations in our model. Moreover, in our manuscript, we applied the non-commutating Hermitian Lindblad operators to every site, which may also lead to relaxation in open quantum systems. However, the phenomenon that a specific initial product state exhibits much longer time periodic revivals than other initial states is clearly seen for our constructed Liouvillian.

We thank the referee for pointing out this important theorem. We will investigate the theorem in detail for future studies of QMBS related to non-Hermiticity, and we have highlighted these points in the section “Summary and Discussion” of the revised manuscript.

List of changes:

  1. We have commented on the jump operators we constructed in relation to experimental realizations in Section 3.5, and emphasized experimentally realizable Lindblad operators in the section “Summary and Discussion”.
  2. We have highlighted the research direction on the eigenmodes of the Liouvillian with purely imaginary eigenvalues from the perspective of the non-Hermitian QMBS.

---

## Round 1 · Referee Report · Anonymous (Referee 2) · 2022-11-27

Strengths

1) In principle, this is an interesting model to explore.

2) Manuscript is well-written, presentation is clear.

Weaknesses

1) Trying to make a connection between properties of excited states such as quantum scars, level statistics to the ground state phases seems to be fundamentally misguided.

2) Does not provide an adequate review of the basic concepts in non-Hermitian systems.

3) Lacks direct experimental motivation. The jump operators that connect this to the Lindblad master equation are highly artificial.

Report

This manuscript contains results on the extension of celebrated quantum many-body scars of the PXP model to an appropriate non-Hermitian generalization of the PXP Hamiltonian. They use standard exact diagonalization to study the properties of this generalized model in the presence of non-Hermiticity and real magnetic field. First, they study various diagnostics in the "ground state" of the model, and they map out the "phase diagram". Subsequently, in each of the ground state phases, they study the fate of quantum many-body scars and revivals, which are known to exist in the PXP model, and they claim that the scars persist in some phases whereas they appear to disappear in some others. In addition, they study the real and complex energy level statistics in each of these phases and find a variety of results.

In principle, exploring the fate of scars under non-Hermiticity is an interesting direction for exploration, given that a community interested in the physics of non-Hermitian Hamiltonians has developed. However, I think there are some fundamental problems in the manuscript, here are some of my major remarks:

1) The way the discussion of scars is presented seems to imply that scarring and ground state properties are correlated, i.e., the ground state phases are also somehow dynamical phases for the special initial state. This is a strong claim, since the definition of a phase implicitly assumes some robustness under perturbations, and other known examples of quantum scars do not seem to have such robustness. However, none of the data the authors present support such a claim. To claim that quantum scars survive in a given phase, it is necessary to show that there a ``sudden" transition of the quantum scar properties at the phase boundaries shown in Fig. 1. Observing a particular behavior for a fixed system size at some isolated points in the phase diagram, as shown in Figs. 3 and 4, is not enough to claim that the behaviour of scars is the same throughout the same phase. Even if the scars exhibit a weak crossover from one behavior to the other at the phase boundaries, it should simply be called a crossover and not a transition unless there is a specific order parameter and scaling collapse in the thermodynamic limit.

2) The above comment also extends to level statistics. The authors seem to claim that the level statistics is correlated with the ground state phases, as shown in Fig. 5. Again, observing a particular behavior at one point in the phase is not sufficient to make the connection between the ground state phase and level statistics. If the authors would really like to claim that, perhaps they can show the level statistics ratios everywhere in the phase diagram of Fig.~1, perhaps in the form of a heat map? To emphasize how strong the claim is -- the authors seem to suggest that the equilibrium phases of matter this model are also non-equilibrium phases of matter, which is mostly unheard of in quantum many-body systems.

3) In probing the ground state physics, the authors assume that several standard concepts in Hermitian equilibrium physics straightforwardly extend to non-Hermitian ground states. First, it is not obvious how ground states are defined in non-Hermitian systems given that the complex spectra, and the authors don't seem to state this clearly anywhere. Second, it is not obvious how and why concepts such as "central charge", "fidelity susceptibility" should generalize -- many of the references that the authors point to seem to be those of standard Hermitian physics.

For these reasons, I do not recommend the publication of the current version of the manuscript in SciPost.

Requested changes

1) The authors should not attempt to make the connection between ground state phase properties and the excited state properties of quantum many-body scars and level statistics without sufficient evidence. If they wish to study both properties, they can reorganize the paper into two disconnected sections -- one focusing on ground states and the other focusing on excited states.

2) The definition of the ground state in non-Hermitian systems needs to be clarified somewhere. It would also be good to briefly review terminologies from the study of non-Hermitian Hamiltonians, including simple concepts such as "overlaps", "bi-orthogonality", "exceptional points", different types of entanglement entropy, etc. Further, when one discusses standard concepts such as "critical exponents", "central charge", "susceptibility", etc., it would help to clarify what these mean and how they are connected to their Hermitian counterparts, which readers would be familiar with.

3) In the introduction, the authors have a sentence "However, whether weak ergodicity breaking exists in the presence of non-Hermiticity is elusive both theoretically and experimentally." The theoretical aspect is not true -- there are papers that explicitly point out exact quantum many-body scars in non-Hermitian Hamiltonians, e.g., https://arxiv.org/pdf/2106.10300.pdf.

  • validity: low
  • significance: ok
  • originality: ok
  • clarity: high
  • formatting: perfect
  • grammar: excellent

Author:  Qianqian Chen  on 2023-02-28  [id 3411]

(in reply to Report 2 on 2022-11-27)

Response to Anonymous Report 2 on 2022-11-27 (Invited Report)

We are immensely grateful to the referee for reviewing our manuscript and providing us with positive appraisals and helpful suggestions. We have accepted all of the requested changes and revised the manuscript accordingly. We hope that these improvements and our point-to-point responses have adequately addressed the referee's concerns.

## Report

Referee’s comment:

In principle, exploring the fate of scars under non-Hermiticity is an interesting direction for exploration, given that a community interested in the physics of non-Hermitian Hamiltonians has developed. However, I think there are some fundamental problems in the manuscript, here are some of my major remarks:

1) The way the discussion of scars is presented seems to imply that scarring and ground state properties are correlated, i.e., the ground state phases are also somehow dynamical phases for the special initial state. This is a strong claim, since the definition of a phase implicitly assumes some robustness under perturbations, and other known examples of quantum scars do not seem to have such robustness. However, none of the data the authors present support such a claim. To claim that quantum scars survive in a given phase, it is necessary to show that there a sudden" transition of the quantum scar properties at the phase boundaries shown in Fig. 1. Observing a particular behavior for a fixed system size at some isolated points in the phase diagram, as shown in Figs. 3 and 4, is not enough to claim that the behaviour of scars is the same throughout the same phase. Even if the scars exhibit a weak crossover from one behavior to the other at the phase boundaries, it should simply be called a crossover and not a transition unless there is a specific order parameter and scaling collapse in the thermodynamic limit. 2) The above comment also extends to level statistics. The authors seem to claim that the level statistics is correlated with the ground state phases, as shown in Fig. 5. Again, observing a particular behavior at one point in the phase is not sufficient to make the connection between the ground state phase and level statistics. If the authors would really like to claim that, perhaps they can show the level statistics ratios everywhere in the phase diagram of Fig.~1, perhaps in the form of a heat map? To emphasize how strong the claim is -- the authors seem to suggest that the equilibrium phases of matter this model are also non-equilibrium phases of matter, which is mostly unheard of in quantum many-body systems.

Our reply (1): Though there are other works indicating that scarring/excited-state and ground-state properties are associated in Hermitian systems [e.g., Quantum 3, 159 (2019); Phys. Rev. B 105, 125123 (2022); arXiv:2210.17032; arXiv:2301.03631], it is indeed not our intention to imply this in our manuscript. We are sorry for our unclear presentation on this issue, and we further distinguish scarring/excited-state and ground-state properties in the revised manuscript following the referee’s helpful comments.

Referee’s comment: 3) In probing the ground state physics, the authors assume that several standard concepts in Hermitian equilibrium physics straightforwardly extend to non-Hermitian ground states. First, it is not obvious how ground states are defined in non-Hermitian systems given that the complex spectra, and the authors don't seem to state this clearly anywhere.

Our reply (2): The referee raised important issues in non-Hermitian quantum mechanics. Indeed, in our manuscript, we have defined what “ground states” means in our system immediately after this terminology is used for the first time, i.e., in the first paragraph in the “Results” section with the sentence “we define the ground state as the state with the smallest real eigenenergy like in Hermitian systems unless stated otherwise”. In the revised manuscript, we have more fully clarified the definition of ground states in the non-Hermitian system.

In this response, we would like to briefly recapitulate the definition of “ground states” in our non-Hermitian system. Since our Hamiltonian has parity-time (PT) symmetry, the whole spectrum is real in the PT symmetry unbroken regime, and these are the regimes that we are most interested in. However, no matter whether the energy spectra are real or complex, we define the ground state by the smallest real eigenenergy, which is consistent with the Hermitian limit (i.e., $\gamma=0$ in Fig.1). When the complex energy spectra appear, we still defined the ground state by the smallest real part of eigenenergy to make our definition consistent.

Referee’s comment: Second, it is not obvious how and why concepts such as "central charge", "fidelity susceptibility" should generalize -- many of the references that the authors point to seem to be those of standard Hermitian physics.

Our reply (3): As for the references, we cited some corresponding references of the standard Hermitian physics in our manuscript since we compare our findings with those of the Hermitian model. And we apologize for only citing the relevant references of non-Hermiticity (such as Ref. [79-81,83-85,91-93]) without making more precise explanations of why concepts such as "central charge", and "fidelity susceptibility" should be generalized. Here, we would give some more explanations:

  1. Central charge: in our non-Hermitian model, the "central charge" is extracted from the generic entanglement entropies (EE) in terms of the biorthogonal basis. Such generalized EE can be reduced to the traditional EE in the Hermitian limit while capturing the necessary characteristics in non-Hermitian critical systems. To be more specific, constructing physical quantities in terms of the biorthogonal basis is widely accepted in the non-Hermitian community and generally adopted in most literature [e.g., see review article Ref. [79] and also Ref. [80,81,91], etc., in our manuscript]. Such formalism can keep consistency with quantum mechanics to a large extent, since basic concepts and principles like probability rules, pure and mixed states, time evolutions, etc., can be well-defined. As we known, a Hermitian Hamiltonian has the orthogonal relation between its eigenstates, while the non-Hermitian Hamiltonian has $\langle \psi_{R,n}|\psi_{R,m}\rangle\neq \delta_{n,m}$ and $\langle \psi_{L,n}|\psi_{L,n}\rangle\neq \delta_{n,m}$ for most eigenstates. Instead, the non-Hermitian Hamiltonian has orthogonality and completeness between left and right eigenvectors, i.e., the biorthonormal relation $\langle \psi_{L,n}|\psi_{R,m}\rangle=\delta_{n,m}$ and $\sum_n|\psi_{R,n}\rangle\langle \psi_{L,n}|=1$. Notably, a non-Hermitian Hamiltonian admits a spectral decomposition as $H=\sum_nE_n|\psi_{R,n}\rangle\langle \psi_{L,n}|$. Therefore, it is natural for us to use a generalized definition of the density matrix $\rho_n=|\psi_{R,n}\rangle\langle\psi_{L,n}|$ and a generalized reduced density matrix

    $$\rho_A=\mathrm{Tr}_\bar A\rho,$$
    with $\bar A$ denoting the complementary part of the subsystem $A$ with a subsystem size $L_A$ . Moreover, it was shown in various critical non-Hermitian systems that such generalized EE gives the expected scaling as a function of the subsystem size $L_A$ similar to the Hermitian case, $S_\mathrm{vN}\sim \frac{c}{3}\ln(\sin\frac{\pi L_A}{N})$ [see Phys. Rev. Lett. 119, 040601 (2017); SciPost Phys. 12, 194 (2022) (Ref. [93] in our manuscript), etc.].

  2. Fidelity susceptibility: the definition of fidelity in the Hermitian system is defined as $\mathcal F_h(|\psi\rangle,|\varphi\rangle)=\langle \psi|\varphi\rangle \langle \varphi|\psi\rangle$ or the square root of $\mathcal F_h$ . Since a complete framework involves the biorthogonal basis on the non-Hermitian Hamiltonian, one of the natural generalizations on the fidelity $\mathcal F_h$ is adopted as

    $$ \mathcal F(\lambda ,\delta \lambda)=\langle \psi_L(\lambda)|\psi_R(\lambda+\delta\lambda)\rangle\langle \psi_L(\lambda+ \delta\lambda)|\psi_R(\lambda)\rangle $$
    It reduces back to the standard fidelity in the Hermitian case. Since the fidelity $\mathcal F=1-\chi\delta\lambda^2+\mathcal O(\delta\lambda^3)$ for non-Hermitian systems with small $\delta$ , one may approximate the fidelity susceptibility $\chi$ as
    $$ \chi≈\frac{1-\mathcal F}{\delta\lambda^2} $$
    as Eq. (8) shows in our manuscript. For a Hermitian system, the fidelity susceptibility $\chi$ always stays positive due to $0\leq \mathcal F\leq 1$. By contrast, the divergence of $\chi$ towards negative infinity implies EPs. See more detail about the generalized fidelity susceptibility for the non-Hermitian system in references such as Ref. [85].

Following the referee’s constructive suggestions, in the revised manuscript, we have supplemented these explanations of the generalized physical concepts after our mentions of them.

Requested changes

Referee’s comment: 1) The authors should not attempt to make the connection between ground state phase properties and the excited state properties of quantum many-body scars and level statistics without sufficient evidence. If they wish to study both properties, they can reorganize the paper into two disconnected sections -- one focusing on ground states and the other focusing on excited states.

Our reply (4): As the titles of the subsections in the section “Results” indicate, we divided the results of “non-Hermitian quantum criticality” and other properties of the non-Hermitian scars, i.e., “The periodic revivals and overlaps of specific states”, “Entanglement entropy and physical observables”, and “Energy level statistics”, into separate subsections. In the revised manuscript, we have further improved our presentation in each subsection to clarify disconnectedly the properties of ground and excited states according to the referee’s constructive suggestion. Moreover, we remove the description of excited properties (i.e., QMBS) from the ground-state phase diagram to avoid misunderstanding.

Referee’s comment: 2) The definition of the ground state in non-Hermitian systems needs to be clarified somewhere. It would also be good to briefly review terminologies from the study of non-Hermitian Hamiltonians, including simple concepts such as "overlaps", "bi-orthogonality", "exceptional points", different types of entanglement entropy, etc. Further, when one discusses standard concepts such as "critical exponents", "central charge", "susceptibility", etc., it would help to clarify what these mean and how they are connected to their Hermitian counterparts, which readers would be familiar with.

Our reply (5): Motivated by this comment, we have improved the presentation in our manuscript to further clarify the definition of the “ground state” in section 3.1, “overlaps” in sections 3.2, 3.3 and the caption of Fig. 4, “bi-orthogonality” in section 2, "exceptional points" in section 1 and 3.1, different types of entanglement entropy in section 3.3., etc.

Additionally, in the revised manuscript, in section 3.1, we have clarified the definitions of "critical exponents" and "central charge" and compared them with the Hermitian counterparts with the conclusion that the central charge is in very good agreement with an Ising universality class at the phase transition. As for "susceptibility", in section 3.1 of the revised manuscript, we have clarified more on its definition and highlighted that the negative divergence of the fidelity susceptibility in our model implies exceptional points, which are impossible to exist in any of standard Hermitian systems.

Some illustrations of these concepts can also be found in Reply (2-3).

Referee’s comment: 3) In the introduction, the authors have a sentence "However, whether weak ergodicity breaking exists in the presence of non-Hermiticity is elusive both theoretically and experimentally." The theoretical aspect is not true -- there are papers that explicitly point out exact quantum many-body scars in non-Hermitian Hamiltonians, e.g., https://arxiv.org/pdf/2106.10300.pdf.

Our reply (6): In the revised manuscript, we have improved the presentation to mention this important work. In detail, we have changed the sentence to “However, there are few studies \cite{Pakrouski2021} regarding whether the weak ergodicity breaking exists in the presence of non-Hermiticity both theoretically and experimentally, especially in the non-perturbative regime.

List of changes:

  1. We have distinguished scarring/excited-state and ground-state properties in the revised manuscript. For instance, we have deleted the words “non-Hermitian QMBS” in Fig. 1, completely disconnecting the ground-state diagram from the properties of excited states.
  2. We have replaced the statistics of energy level spacing at $h=-0.3, \gamma=0.5$ with that at $h=0, \gamma=0.1$ in Fig. 5a to better compare with Fig. 4a with $h=0, \gamma=0.1$.
  3. We have improved the presentation in our manuscript to further clarify the definition of the “ground state” in section 3.1, “overlaps” in sections 3.2, 3.3 and the caption of Fig. 4, “bi-orthogonality” in section 2, "exceptional points" in section 1 and 3.1, different types of entanglement entropy in section 3.3.
  4. In section 3.1, we have clarified the definitions of "critical exponents" and "central charge" and compared them with the Hermitian counterparts with the conclusion that the central charge is in very good agreement with an Ising universality class at the phase transition. As for "susceptibility", in section 3.1 of the revised manuscript, we have clarified more on its definition and highlighted that the negative divergence of the fidelity susceptibility in our model implies exceptional points, which are impossible to exist in any of the standard Hermitian systems.
  5. In the section “introduction” of the revised manuscript, we have modified the sentence regarding the weak ergodicity breaking in non-Hermitian systems.

---

## Editorial Decision

resubmitted